# Integrated Machine Vision System for Evaluating Hole Expansion Ratio of Advanced High-Strength Steels

**DOI:** 10.3390/ma15020553

**Published:** 2022-01-12

**Authors:** Jaehoon Park, Chanhee Won, Hye-Jin Lee, Jonghun Yoon

**Affiliations:** 1Department of Mechanical Design Engineering, Hanyang University, 222, Wangsimni-ro, Seongdong-gu, Seoul 04763, Korea; jayhoon20@gmail.com; 2BK21 FOUR ERICA-ACE Center, Department of Mechanical Engineering, Hanyang University, 55, Hanyangdaehak-ro, Sangnok-gu, Ansan 15588, Gyeonggi, Korea; 3Digital Transformation R&D Department, Korea Institute of Industrial Technology, 143, Hanggaul-ro, Sangrok-gu, Ansan 15588, Gyeonggi, Korea; chan2@kitech.re.kr (C.W.); naltl@kitech.re.kr (H.-J.L.); 4AIDICOME Inc., 55, Hanyangdaehak-ro, Sangnok-gu, Ansan 15588, Gyeonggi, Korea

**Keywords:** advanced high-strength steel (AHSS), edge crack, hole expansion test, hole expansion ratio (HER), computer vision, punch load, acoustic emission

## Abstract

In this paper, we propose a new method to estimate the hole expansion ratio (HER) using an integrated analysis system. To precisely measure the HER, three kinds of analysis methods (computer vision, punch load, and acoustic emission) were utilized to detect edge cracks during a hole expansion test. Cracks can be recognized by employing both computer vision and a punch load analysis system to determine the moment of crack initiation. However, the acoustic emission analysis system has difficulty detecting the instant of crack appearance since the magnitude of the audio signal is drowned out by noise from the press, which interrupts the differentiation of crack configuration. To enhance the accuracy for determining the HER, an integrated analysis system that combines computer vision with punch load analysis, and improves on the shortcomings of each analysis system, is newly suggested.

## 1. Introduction

Advanced high-strength steels (AHSSs) have been widely applied to body-in-white (BIW) structures for both weight reduction and crass worthiness because they exhibit higher tensile strength compared with conventional mild steels [1,2,3]. However, unexpected edge fractures occur during the metal forming process due to low sheared edge cracking resistance, which prevents their use in automotive body applications [4,5]. Predicting edge cracks utilizing numerical simulations with a forming limit diagram is difficult since strain at the edge is less than that of the forming limit curves, which gives rise to enormous issues in manufacturing BIW structures with AHSS sheets [6,7].

To predict the edge cracks of AHSS sheets during the stamping and flanging process, hole punching and expansion tests (which are common methods to evaluate edge stretchability) are utilized in industries for estimating the quantitative amounts of edge fracture resistance [8,9]. The hole expanding process, which is defined in the ISO 16630-2009 standard, consists of punching a hole on a test specimen and using a conical punch with 60 degrees at the top which moves upward to extend through the thickness of the AHSS sheets [10]. The hole expansion ratio (HER) is an index to determine the flangeability of a material, which is calculated using the ratio of the initial inner diameter to the final inner diameter, where the final crack occurs on the sheared edge surface [11].

Conventionally, cracks from the hole expansion test are identified by the human eye, which leads to inconsistent measurements and increases the inaccuracy of the HER evaluation. Furthermore, edge cracks of AHSS occur at irregular times for each trial of the hole expansion test, which results in the deviation of the experimental results. It is difficult to decide the exact moment of the edge crack, since the crack propagation is rapidly proceeding. There are several approaches to examine the hole expansion test and capture the moment when the onset of cracking occurs. The period of crack occurrence and propagation is too rapid to follow with the human eye. Therefore, a high-resolution live camera with a sufficient frame rate has been used for capturing the images during the hole expansion test, which helps inspectors to determine the exact time of edge cracking.

Extensive studies related to the hole expansion test have been conducted using numerical simulations and experimental methods to evaluate the HER precisely. Chiriac et al. [12] suggested a digital recording and measurement method to determine the HER for three dual phase steels, which are DP590, DP780, and DP980, with various thicknesses. They utilized a high-resolution camera and fast type of data transfer system to ensure the test results for a variety of AHSS sheets and evaluated the HER values with different test conditions. However, the number of conducted tests and the evaluation of the standard deviation are not specified to validate the experimental results. Behrens et al. [13] carried out a two-track analysis to compare the precision of each HER measurement. They conducted hole expansion tests of DP780 sheets utilizing both a vision camera and an acoustic emission sensor. They analyzed the punch stroke at the edge failure for both methods to evaluate the capability of detecting cracks for each of them. Chen et al. [14] analyzed the punched hole edge morphology of AHSSs including DP780, DP980, and MS1180. They constructed the optical measurement system, which consists of a high-performance stereo optical microscope, to observe the full crack through the thickness. They also conducted a hole expansion test three times for each AHSS to validate the experimental results. Nevertheless, the measurement methods are not automatically able to analyze the fracture morphology. Wang et al. [15] utilized a 3D digital image correlation technique to measure the HER of 1.6 mm-thick DP780 sheets and compared the results with the measurement by the caliper. However, both measurement results show discrepancies, with difficulty in determining the HER accurately. Choi et al. [16] developed a real-time crack detection system with an image processing algorithm in the hole expansion test to precisely capture the images of crack identification. They utilized a vision camera and light-emitting diode (LED) light source to enhance the resolution and proposed a versatile crack inspection algorithm.

Previous studies have mainly focused on making use of a computer vision-based measurement system in the hole expansion test. However, results have depended on the experimental circumstances such as the intensity of the light source and the cleanliness of the test specimen, which is strongly related to the recognition of the crack configuration from the computer vision-based measurement systems. Therefore, a novel technique to evaluate the HER with minimum prediction error is needed. There are some other research approaches to perceive crack appearance during a hole expansion test.

Dünckelmeyer et al. [17] investigated the punch load–displacement to detect the onset of cracks during the hole expansion test. They found a correlation between the punch load drop and the damage to the punched hole edge. Krempaszky et al. [18] analyzed the onset and propagation of the crack by measuring the maximum punch force and the displacement with variation of the conical punch angle and the initial hole diameter in the hole expansion test of CP800 and DP800 sheets. Panich et al. [19] conducted the hole expansion test with the conical punch and examined the tendency of the punch load with respect to the punch displacement. They discovered the load drop after the crack occurrence. Barnwal et al. [20] recently conducted hole expansion tests of DP980 and TRIP1180 sheets utilizing the flat bottom punch and examined the relationship between the fracture morphology and the HER. In this case, the punch load dropped after the initial crack started in both experiments and simulations of the hole expansion test. Anderson et al. [21] suggested a finite element damage model of the hole expansion test of DP780 sheets. They performed both experiments and simulations for comparison and found that the punch force drops after the crack appears.

Recently, numerous studies concerning hole expansion tests have been conducted with various AHSS sheets and hole edge conditions. Paul [22] carried out hole expansion tests with various kinds of AHSS sheets to evaluate the edge formability, but their HER measurement results have large deviations due to the inconsistent HER evaluation. Santos et al. [23] developed a test machine for the hole expansion test to evaluate the HER in terms of various thicknesses. They utilized real-time image acquisition from a video camera, but there was a large deviation in the HER results due to the different material conditions such as light reflection. Balisetty et al. [24] evaluated the HER of DP600 steel with respect to numerous hole processing conditions such as shearing, milling, and with an electrical discharge machine (EDM). They constructed a camera and light source to identify the thickness of the crack initiation, but they did not analyze the procedure of the crack initiation and propagation, which led to large deviations in the HER, although the same material and hole machining methods were applied.

This paper suggests a precise method to predict the edge crack identification in hole expansion tests of AHSS such as DP980, TRIP1180, and MART1500. To determine the exact moment of crack appearance, methods for computer vision, punch load–displacement, and acoustic analysis have been developed to mitigate the issues associated with the existing measurement. These measurements have been performed to evaluate the HER and were compared with that of ground-truth data to determine whether they have an influence on edge stretchability. Quantitative analyses have been conducted to determine the appropriate methods for maximizing the prediction and accuracy of HER.

Choi et al. [16] focused on developing the machine vision system and detecting algorithm for automatic crack detection in hole expansion tests, but it is limited to analyzing the error of incorrect crack identification when the light reflection of the tested material is changed, since the accuracy of the machine vision system is sensitive to changes of light reflection. To overcome this limitation, this paper newly integrated a punch load analysis system into a machine vision system, which makes it possible to detect crack occurrence, although the machine vision system cannot precisely detect the crack due to the excessive light deflection, etc.

## 2. Experimental Procedure

### 2.1. Test Materials

To evaluate the material properties of AHSSs, a variety of steels which exhibit greater than 1.0 GPa tensile strength, namely DP980, TRIP1180, and MART1500, have been selected as test materials. The sheet thickness was 1.2 mm. The tensile test specimens were machined based on ASTM-E8, which are applied to the quasi-static tensile tests at the strain rate of 0.001/s at room temperature. Figure 1 demonstrates the engineering stress–strain curves of tested materials, which shows that MART1500 had the highest tensile strength and lowest fracture ductility compared with the other two materials. Detailed material property information is listed in Table 1.

### 2.2. Hole Expansion Test

Hole expansion tests were conducted according to the ISO 16630-2009 standard to estimate the quantitative amount of edge stretchability. Due to the high contact pressure between the sheet metal and die set, a 100-ton servo press (CHONG RO SCIENTIFIC CO. LTD, Seoul, Korea) has been constructed to endure an enormous reaction force during the hole expansion test, as shown in Figure 2.

Test specimens with a hole diameter of 10 mm at the center were prepared by a hole punching process with a clearance of 12% with respect to the sheet metal thickness. The sample was placed on the punch–die set with the burr facing upward. A conical (cone angle of 60°) expanding tool moved upward at a speed of 0.1 mm/s under a blank holding force of 25 tons, which was exerted by the die set. The test was stopped when a through-thickness crack appeared at the hole edge. The whole process was recorded using the vision camera.

Measurement procedure of the HER is depicted in Figure 3. Firstly, the computer vision analyzes the HER testing images using the image processing algorithm, which consists of image binarization, blob detection, background deletion, region of interest (ROI) selection, and image linearization. If the vision analysis identifies the edge crack, the test is stopped and the HER is automatically calculated. However, if the crack cannot be identified using the machine vision system, additional systems such as the punch load and acoustic emission analysis are conducted to identify the crack to evaluate the HER.

To define the diameter of the test specimen, an arbitrary point was chosen inside the circle and two lines which are perpendicular to each other were selected. Two points that pass the two lines are generated, and the other two lines are drawn to determine the center of the circle. The measured radius is the distance between the center and intersection point of the inner circle and a line which passes the arbitrary point. The HER is determined by the initial and final diameter of the hole expansion test, which is the average diameter of the two perpendicular directions without cracks as shown in Figure 4. The value was calculated using Equation (1), where $D_{0}$ and $D_{F}$ refer to the initial and final inner diameters of the test specimen, respectively.
(1)$$\mathrm{HER}\left(\%\right)=\left(\frac{D_{F}-D_{0}}{D_{0}}\right)\times 100$$

Before examining the HER with various types of analyses, a manual measurement was conducted by human visual inspection, which is the conventional method to determine the moment of crack appearance and to evaluate the HER. The test using the manual measurement was stopped when a human inspector announced the occurrence of a through-thickness crack. To secure the reliability of the measurement, five human inspectors were mobilized for estimating the HER. The average value and the standard deviation of the HER were calculated by five manual measurements with the human eye.

#### 2.2.1. Machine Vision Analysis for Determining HER

Once the hole expansion test began, a grayscale 5 M (2592-1944) pixel CCD vision camera (10 frames per second, The Imaging Source Asia Co. LTD, New Taipei, Taiwan) was used to take the images in real-time. A bitelecentric lens was attached in front of the camera to minimize the error caused by the relative distance between the camera and the test specimen while the images were acquired. A light source system consisting of 36 LEDs and a reflective plate was placed between the camera and the die set to improve the image quality while the cracks appeared and propagated through the thickness during the experiments, as shown in Figure 5. The test was stopped as the through-thickness cracks progressed. The initial and final inner diameter were measured in the first and the last images in a series of hole expansion tests, and the value of the HER was automatically calculated by a connected computer.

An image processing algorithm [16] is utilized for precisely measuring the HER, as illustrated in Figure 6. It consists of image binarization, blob detection, background deletion, ROI selection, image linearization, and crack identification. Image binarization was used to classify the captured images into white and black images to determine the through-thickness crack from the input images. Blob detection was utilized for establishing a group of white pixels in the binarized image by combining the adjacent pixels and removing the inessential areas from the binarized images in the crack identification to determine the region of interest (ROI). To pick out the appropriate ROI, the background, which is the largest unnecessary region, should be deleted from the blob group. After the ROI was selected, image linearization was conducted to efficiently detect the cracks.

#### 2.2.2. Punch Load Analysis for Determining HER

In order to achieve the precise punch load and displacement data, a load cell (BONGSHIN LOADCELL Co. LTD, Osan, Korea) with 100 tons of maximum force and 0.1 kg of resolution is applied during the HER experiments, and the data acquisition system collected 32 punch load and displacement data per second in real-time measurement. The punch load rapidly increased until the first appearance of a through-thickness crack. The increasing tendency slows down while the crack propagates. At last, the punch load reached the peak and then dropped, exceeding the threshold of crack resistance. The evaluation standard of edge failure was 0.2% for the maximum punch load. This is the moment when the crack propagation was finalized by observation of the live video. The related process is demonstrated in Figure 7.

Figure 8 demonstrates the real-time captured images which correspond to the punch load data. It is shown that the sheared surface gradually flipped over before 10 mm of punch displacement during the hole expansion, and the onset of the microcrack is observed before reaching the maximum punch load. Then, the microcracks propagate rapidly through the thickness direction within 0.3 mm of the punch displacement.

#### 2.2.3. Acoustic Analysis for Determining HER

The crack detection technique examining the acoustic signal is utilized in determining fatigue cracks. Similarly, the acoustic system consisted of a sensitive microphone and a FFT analyzer (ATS Technology, Inc., Seoul, Korea). A microphone with 1 dB resolution was utilized to detect the audio signal, while the onset of the crack occurred and proceeded during the hole expansion test. A four-channel FFT analyzer received the acoustic signal from the microphone and converted the time domain response to the frequency domain.

## 3. Results and Discussion

Three kinds of prediction systems, including vision, punch load, and an acoustic analysis system, were used to precisely measure the HER. The main system for evaluating the HER was the computer vision system. However, it cannot be used in the cases of excessive light reflection on the punched surface, which interferes the crack detection even though the same material and test conditions are used, as illustrated in Figure 9. Therefore, subsystems for punch load–displacement and acoustic analysis have been developed to prevent the overestimation of the HER by utilizing a vision analysis system.

### 3.1. Machine Vision Analysis System in HER Test

The experimental result of the machine vision analysis system is demonstrated in Table 2.

It showed that the values of the average HER followed the order of MART1500, DP980, TRIP1180, and were 16.6%, 18.6%, and 23.6%, respectively. The standard deviation also showed the same trend. Here, higher values of HER revealed more measurement error. Although MART1500 exhibited the highest tensile strength among the materials, it revealed the lowest edge stretchability, which limits its use in automotive BIW structures. Compared with manual measurement results, as shown in Table 3, the standard deviation decreases prominently due to automated crack detection, even though it shows a slightly lower average HER value. This is because it avoids the personal error from human visual inspection.

### 3.2. Punch Load Analysis System in HER Test

The experimental result of the punch analysis system is demonstrated in Table 4. It showed that the values of the average HER for DP980, TRIP1180, and MART1500 were 18.5%, 23.5%, and 16.5%, respectively, which is the same trend as the previous results of the computer vision analysis system. The average HER value showed a slight increase, which means that the load cell was inferior to the vision camera with respect to capturing the moment when the cracking occurs. However, the standard deviation decreased significantly, which shows that the punch load analysis system does not consider the test conditions and circumstances as compared with the vision analysis system.

### 3.3. Acoustic Analysis System in HER Test

The experimental result of the acoustic measurements was taken at four points: the beginning of the test, the onset of the crack, the propagation of the crack, and the edge fracture. The measurement results are displayed in Figure 10.

However, the intensity of the audio emission does not show a dramatic difference between points. This is because of the noise from the press, which was louder than the audio emission of the cracking. Noise from the cracking was drowned out during the hole expansion test, which makes it difficult to estimate the HER by utilizing the acoustic analysis system.

### 3.4. Integrated Analysis System in HER Test

To address the shortcomings of each analysis system, a combined analysis system has been proposed. The measurements were mainly conducted with the machine vision analysis system. In the case of overestimating the HER, the punch load analysis system was started, and the test was stopped at the standard punch load drop.

This integrated analysis system provides a ground-breaking precise measurement of the HER. The structure of it is demonstrated in Figure 11, and the experimental result is shown in Table 5.

The computer vision analysis system has the disadvantage of a relatively low precision because of some abnormal cases for high light reflection on the ROI, which prevents precise measurements. Due to the advantages of the punch load analysis system, the results were a more accurate evaluation, proving it to be a superior measurement system for the HER, as shown Figure 12 and Figure 13.

## 4. Conclusions

In this paper, we proposed a novel method to measure the HER using a crack identification system. To evaluate the HER precisely, three analysis methods, including computer vision, hole expanding punch load, and acoustic emission analysis, were conducted. Machine vision and the punch load analysis system could detect the onset of a crack, the propagation of a crack, and the final failure. However, the acoustic analysis system did not recognize the crack progression since the press noise was higher than the sound of the emission for the crack configuration. Therefore, an integrated analysis system consisting of visual analysis with image processing and load drop analysis for hole flanging tools was built. Based on our evaluation of the system, the following conclusions were drawn:The machine vision analyzing system shows the upmost average value of the HER among the other analyzing systems and a lower standard deviation than the manual measurement due to the automated image processing algorithm, which prevents personal error and enables the better precision of the measurement. However, it has some limitations in applying to cases for excessive light reflection on punched surfaces, which interferes with the crack detection.The punch load analyzing system has the least measurement error because it has less sensitive experimental environments compared with the machine vision analysis system. For this reason, it can overcome some extraordinary cases of crack detection when the machine vision analyzing system cannot recognize cracks, which indicates the capability to compensate the disadvantage of the machine vision analyzing system.The integrated analyzing system, which combines the machine vision and punch load analyzing system, fulfils both accurate HER values and lessens measurement uncertainty. Although the punch load analysis system shows the lowest measurement deviation, the amount of close prediction to manual measurement is higher compared with the punch load analysis system, which can replace the previous measurement and analyzing system.Acoustic analysis systems were newly proposed in this paper, but they are not an appropriate method to evaluate the HER. If an advanced microphone and FFT analyzer in a soundproof space is developed in the future, acoustic systems may be suitable for use in hole expansion tests.

## Figures and Tables

**Figure 1 materials-15-00553-f001:**
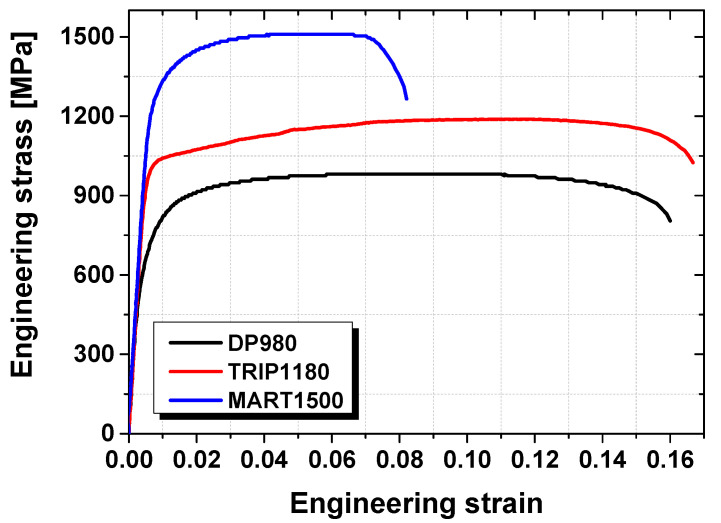
Engineering stress–strain curve of DP980, TRIP1180, and MART1500.

**Figure 2 materials-15-00553-f002:**
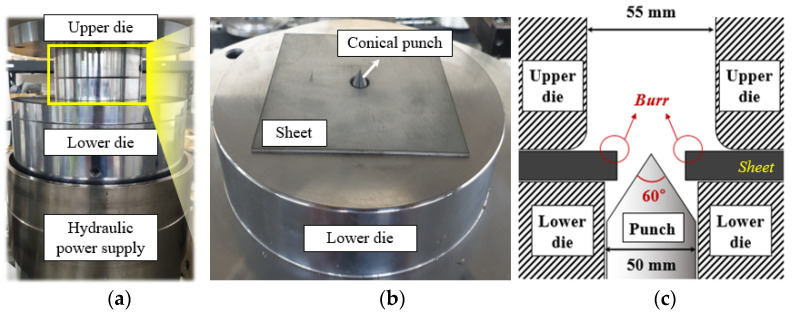
Schematic diagram of hole expansion test set-up: (**a**) test equipment; (**b**) initial set-up of hole expansion test; (**c**) cross-sectional view of hole expansion test.

**Figure 3 materials-15-00553-f003:**
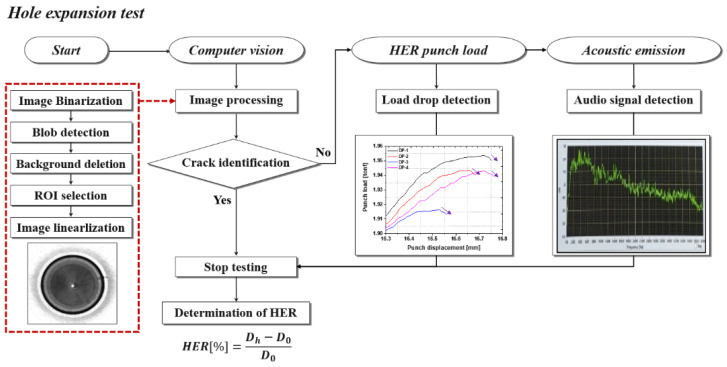
Measurement procedure of HER with various kinds of analysis system.

**Figure 4 materials-15-00553-f004:**
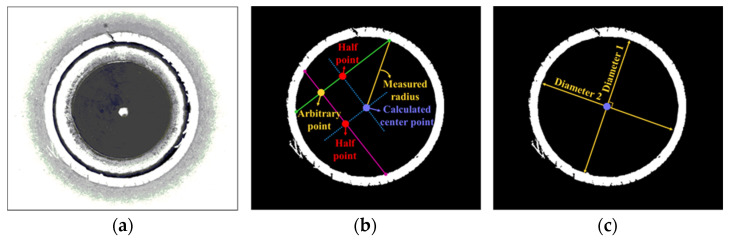
Method for determination of HER: (**a**) captured image from vision camera; (**b**) method for calculating center point at hole; (**c**) measuring standard for defining diameter from hole.

**Figure 5 materials-15-00553-f005:**
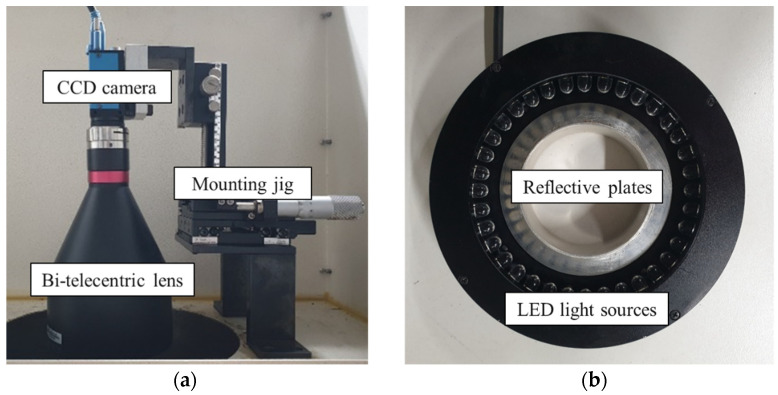
Experimental tools for vision analysis system: (**a**) vision camera with lens; (**b**) light sources and reflective plates.

**Figure 6 materials-15-00553-f006:**
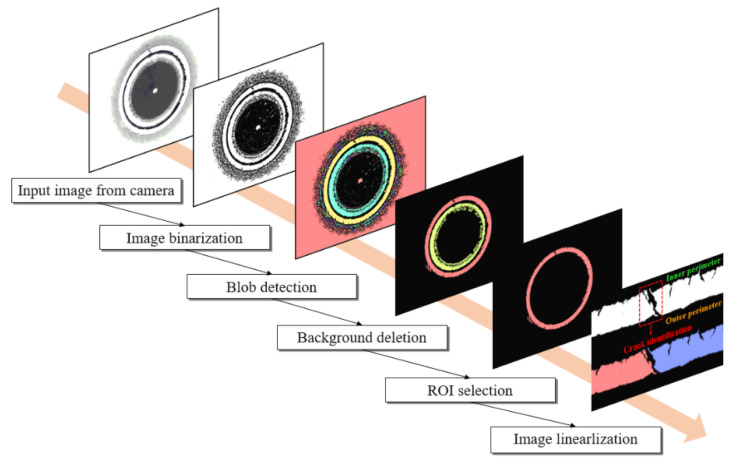
Image processing procedure for crack identification [16].

**Figure 7 materials-15-00553-f007:**
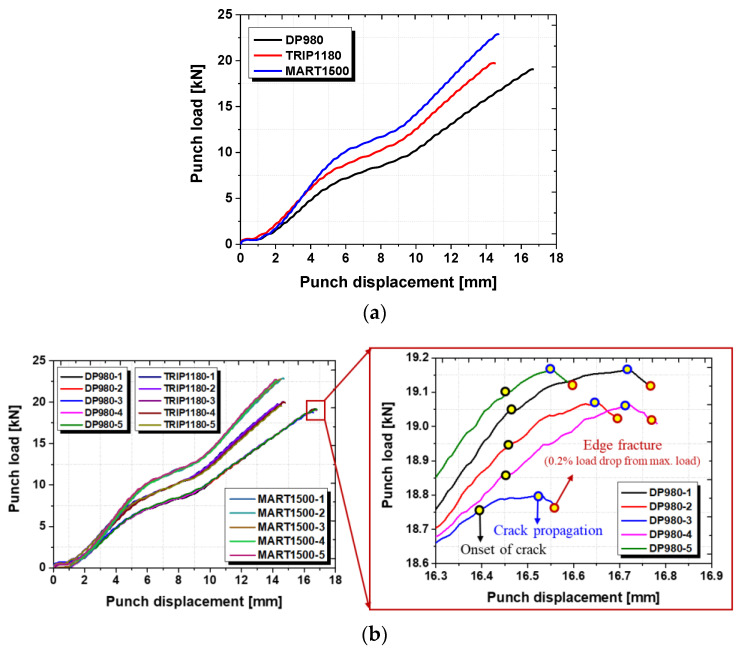
Punch load–displacement analysis with images by live video: (**a**) punch-load displacement curve in terms of AHSS sheets; (**b**) classification of crack formation in punch-load displacement curve; (**c**) enlarged test image for onset of crack, crack propagation and edge fracture.

**Figure 8 materials-15-00553-f008:**
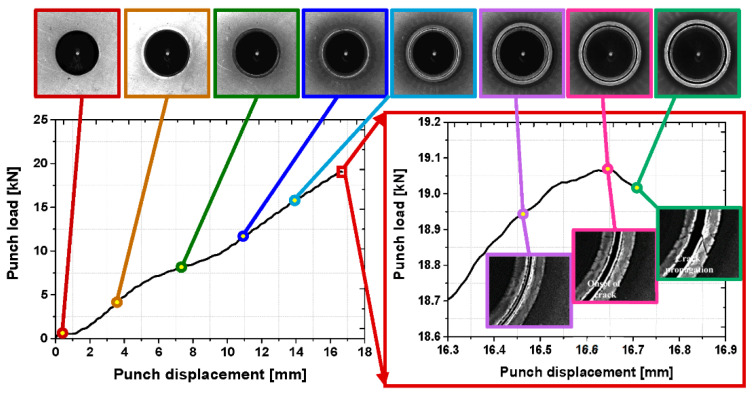
Matched results between test images and punch load data in integrated machine vision system.

**Figure 9 materials-15-00553-f009:**
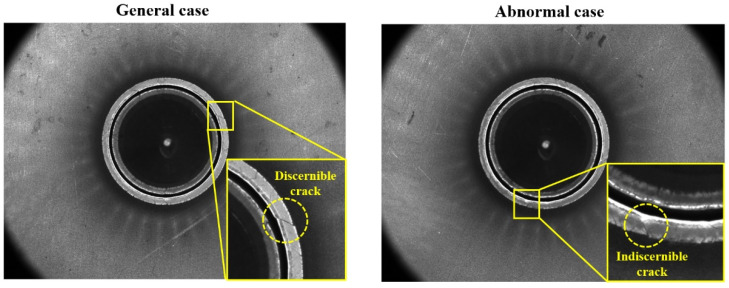
Cases for crack recognition by making use of machine vision analyzing system.

**Figure 10 materials-15-00553-f010:**
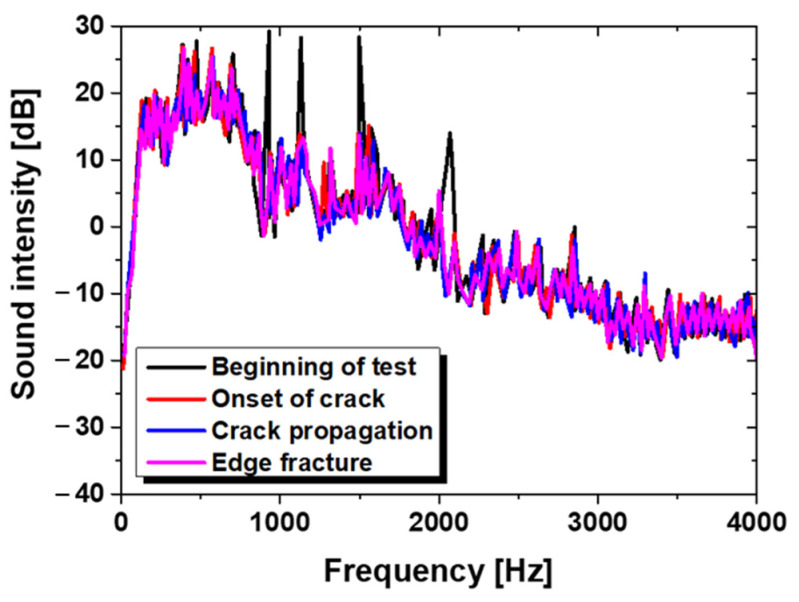
Acoustic measurement and FFT analysis during hole expansion test.

**Figure 11 materials-15-00553-f011:**
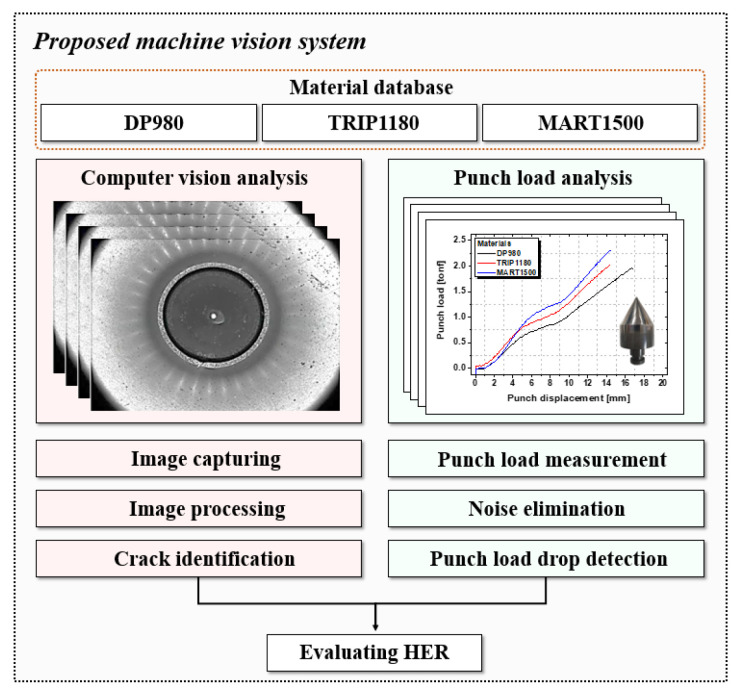
Integrated machine vision system for evaluating HER of AHSS.

**Figure 12 materials-15-00553-f012:**
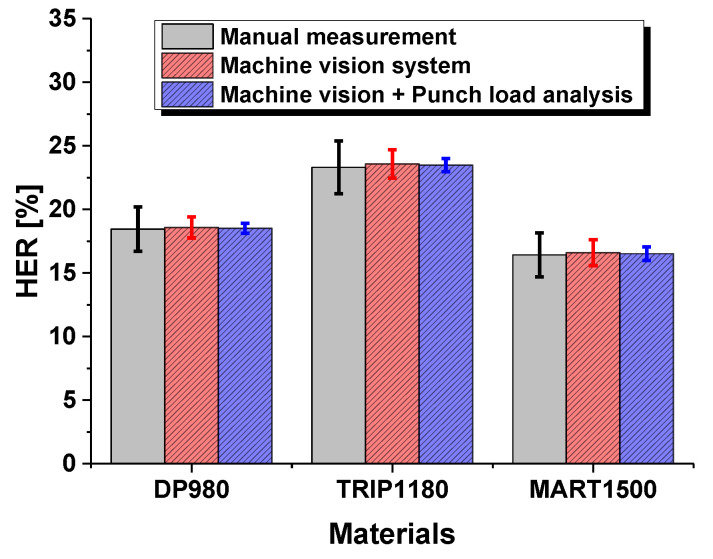
Experimental results by HER evaluation method in hole expansion test.

**Figure 13 materials-15-00553-f013:**
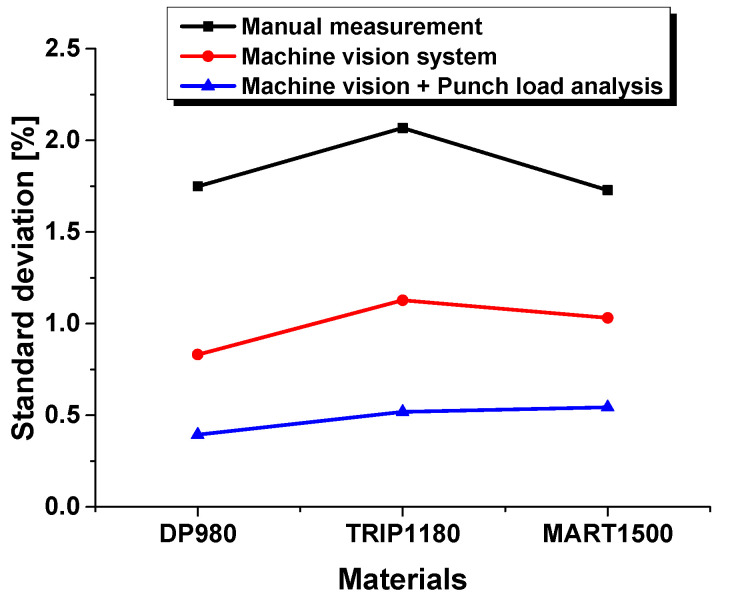
Standard deviation with respect to HER evaluation method in hole expansion test.

**Table 1 materials-15-00553-t001:** Material properties of DP980, TRIP1180, and MART1500.

Material Property	DP980	TRIP1180	MART1500
Yield strength (MPa)	683.4	968.2	1235.1
Tensile strength (MPa)	981.2	1188.7	1509.7
Total elongation (%)	16.0	16.7	8.28
$\mathrm{Flow}\;\mathrm{curves}\;(\bar{\sigma }=K{\bar{\varepsilon }}^{n}$)	$\bar{\sigma }=1381.5{\bar{\varepsilon }}^{0.0718}$	$\bar{\sigma }=1623.2{\bar{\varepsilon }}^{0.128}$	$\bar{\sigma }=2158.1{\bar{\varepsilon }}^{0.106}$

**Table 2 materials-15-00553-t002:** Measured HER by utilizing machine vision system.

Number of Experiment	DP980	TRIP1180	MART1500
Experiment #1	19.5	24.1	17.2
Experiment #2	18.9	23.8	16.8
Experiment #3	17.9	22.9	15.1
Experiment #4	17.3	22.1	16.8
Experiment #5	16.9	21.1	15.3
Experiment #6	18.8	24.4	16.8
Experiment #7	18.6	23.8	14.9
Experiment #8	19.2	23.5	16.4
Experiment #9	18.6	22.3	17.2
Experiment #10	18.4	24.9	16.3
Experiment #11	19.4	21.8	17.2
Experiment #12	18.9	25.3	15.8
Experiment #13	17.5	23.1	17.6
Experiment #14	18.4	23.2	18.4
Experiment #15	19.8	24.6	17.9
Experiment #16	18.2	25.3	16.4
Experiment #17	18.7	24.4	15.3
Experiment #18	19.2	22.9	16.2
Experiment #19	17.5	24.3	15.8
Experiment #20	20.1	23.8	18.6
Average value	18.6	23.6	16.6
Standard deviation	0.83	1.13	1.03

**Table 3 materials-15-00553-t003:** Measured HER by manual measurement.

Number of Experiment	DP980	TRIP1180	MART1500
Experiment #1	20.3	25.6	18.1
Experiment #2	16.8	24.4	16.8
Experiment #3	19.2	22.9	14.3
Experiment #4	17.4	20.3	17.2
Experiment #5	15.1	19.8	13.5
Experiment #6	18.2	21.3	15.2
Experiment #7	19.3	25.8	17.3
Experiment #8	21.2	23.6	15.4
Experiment #9	19.4	19.4	13.9
Experiment #10	16.9	24.3	17.6
Experiment #11	18.1	25.8	18.3
Experiment #12	19.5	23.9	13.8
Experiment #13	18.6	24.6	18.9
Experiment #14	16.9	21.4	18.6
Experiment #15	18.3	25.4	14.8
Experiment #16	20.4	21.7	18.2
Experiment #17	15.9	24.8	15.4
Experiment #18	19.7	24.3	17.3
Experiment #19	21.6	25.6	18.1
Experiment #20	16.2	21.4	15.8
Average value	18.5	23.3	16.4
Standard deviation	1.75	2.07	1.73

**Table 4 materials-15-00553-t004:** Measured HER by applying punch load analysis in machine vision system.

Number of Experiment	DP980	TRIP1180	MART1500
Experiment #1	19.1	24.0	16.7
Experiment #2	18.9	22.6	16.9
Experiment #3	18.2	22.9	16.1
Experiment #4	18.4	23.2	17.0
Experiment #5	18.0	23.7	16.3
Experiment #6	18.2	24.2	15.8
Experiment #7	17.8	24.1	16.4
Experiment #8	18.5	23.8	15.6
Experiment #9	18.1	23.9	16.5
Experiment #10	18.4	23.6	17.3
Experiment #11	18.7	22.4	17.1
Experiment #12	19.3	24.1	16.3
Experiment #13	19.1	23.8	15.6
Experiment #14	18.2	22.9	17.1
Experiment #15	18.6	23.5	17.0
Experiment #16	18.9	23.8	16.4
Experiment #17	18.6	22.9	17.4
Experiment #18	18.2	23.4	16.8
Experiment #19	18.8	23.1	15.8
Experiment #20	18.4	23.8	16.2
Average value	18.5	23.5	16.5
Standard deviation	0.39	0.52	0.54

**Table 5 materials-15-00553-t005:** Measured HER with various analyzing system.

Analyzing System	DP980		TRIP1180		MART1500	
	Average HER	Standard Deviation	Average HER	Standard Deviation	Average HER	Standard Deviation
Manual measurement	18.45	1.749	23.32	2.067	16.43	1.729
Machine vision system	18.59	0.831	23.58	1.128	16.60	1.031
Vision system with punch load	18.52	0.394	23.49	0.519	16.52	0.543

## Data Availability

All the data are available within the manuscript.

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
