# Peer review of "Integrated Machine Vision System for Evaluating Hole Expansion Ratio of Advanced High-Strength Steels"

_materials, 2022, doi:10.3390/ma15020553_

Round 1
Reviewer 1 Report
This paper proposes a measurement method of the hole expansion ratio. The image analysis supported with punch load is proposed. It solves the problem of scattering of the obtained results and the error due to glare in image analysis.
High-strength steel sheets are increasingly used in automobiles and have a low hole expansion rate. The research of the hole expansion ratio is industry valuable due to formability.
However, there are problems such as lack of explanation about the proposed method, lack of novelty, and lack of data.
Therefore, major revisions or additional experiments are required.
1. proposed method is mostly written in reference 16, and this paper is not novel; the novelty is not clearly explained. The problems of Reference 16 should be explained in the introduction, and the novelty of this paper should be clarified.
2. The advantages of the proposed method are shown by comparing the standard deviation. However, the number of experiments is too small. Since the results in standard deviation are similar, many experiments are required.
3. Although it is called hybrid, it is difficult to define the proposed method as a hybrid. It is no more than replacing visual analysis with punch load analysis in case of error generated in visual analysis. That is not a hybrid, as each method is independent. There is no interrelationship.
Instead, the result in Figure 7 is hybrid. The hole expansion ratio is immediately obtained by punch loading, and information on the crack propagation that differs depending on the steel type is obtained from the image analysis.
I think these results are more valuable than the proposed method.
4. The explanation of figure 3 is used to explain the reason for using the servo press, but it is not appropriate.
7, An explanation of figure 4 is needed.
The following is just a comment.
The difference between the average value of the manual measurement and the visual analysis is slight, and the advantages of image analysis are not so much. Instead, I think the advantage of visual and punch load analysis is speed. I think it is effective for tests under a high strain rate.
Author Response
We appreciate your valuable comments. Please see the attachment.

Reviewer 2 Report
The research topic is important and well suited in the journal scope. The results obtained are well interpreted. Moreover, structure of this manuscript is acceptable.
There are some more comments from this reviewer that can help the Authors to improve the quality of the manuscript. Minor revisions are suggested. The detailed comments in the order of appearance in the text are summarized as follows.
When analysing the HER by human visual inspection, when and how is the diameter measured? I am thinking about the springback of the hole
after releasing the punch force
In point 2.2.2, I think it is better to describe the load cell with data about the maximum force and the error
In Fig 7 it is not clear what curves correspond to each material
In Fig 7 i would suggest to define the force units in kN
Are the images and the force value signal recorded in the same temporal scale? I mean, if we could define at what point of the force signal
the image is taken that could give the authors interesting information about the differences between the HRE values
It is not clear enough the hybrid analysis system. When do authors consider that the HRE is overestimated with the vision system? and
when doing this, at the end the punch load analysis defines the HRE, doesn´t it?
In conclusion 2, it says that unch load analysing system reveals the inferior average value of HER compared with machine vision analysing system
but in fact it is the contrary i think
Author Response

(The authors gave the same response as above.)

Reviewer 3 Report
The authors performed very valuable research and provided practical results. The structure of the article is well written. But, I believe that the quality of the manuscript can be improved using the following points:
1- The literature review should be improved by using more papers published recently and describing more details.
2- Related to the "2.1 test materials", it is better to describe the tensile test with more details such as environmental conditions, strain rate, etc.
3- It is laboratory research and very valuable. But it is better to state how many samples have been taken to check the response repeatability? Is there only one test for each case? In this case, there may be various errors, including the operator and ..... then, what is your opinion about the reliability of the result?
Author Response

(The authors gave the same response as above.)

Reviewer 4 Report
A very interesting paper of great practical importance
In this paper, Authors proposed a novel method to measure HER using a crack identification system. To evaluate HER precisely, three analysis methods including computer vision, hole expanding punch load, and acoustic emission analysis were conducted. Machine vision and the punch load analysis system could detect the onset of a crack, propagation of a crack, and final failure. The acoustic analysis system did not recognize crack progression since the press noise was higher than the sound of emission for crack configuration.
The most important conclusions include:
1. Hybrid analysing system which integrates the machine vision and punch load analysing system fulfils both accurate HER values and lessens measurement uncertainty.
2. Acoustic analysis systems were newly proposed in this paper, but they are not an appropriate method to evaluate HER.
3. If an advanced microphone and FFT analyser in a soundproof space is developed in the future, acoustic systems may be suitable for use in hole expansion tests.
Author Response

(The authors gave the same response as above.)

Round 2
Reviewer 1 Report
Thank you for your reply. The quality of the article has remarkably improved. Proposed methods are interesting for the evaluation of high-strength steels.